

# 10,000 years of melt history of the 2015 Renland ice core, East Greenland

Tetsuro Taranczewski[1], Johannes Freitag[1], Olaf Eisen[1,3], Bo Vinther[2], Sonja Wahl[2], and Sepp Kipfstuhl[1]

[1]Alfred Wegener Institute, Helmholtz Centre of Polar and Marine Research, Bremerhaven, Germany
[2]Centre for Ice and Climate, Niels Bohr Institute, University of Copenhagen, Copenhagen, Denmark
[3]Deptartment of Geosciences, University of Bremen, Bremen, Germany

**Correspondence:** Tetsuro Taranczewski (tetsuro.taranczewski@awi.de)

**Abstract.** An ice core drilled in 2015 on the Renland ice cap at the eastern margin of Greenland has been inspected with regard to its melt content. The thickness of a melt layer reflects the temperature level at the time of melt generation. Hence the melt layers are an indicator of past regional summer temperatures in East Greenland, a region where paleoclimate records are sparse. Melt layers have been identified almost along the whole core, resulting in a melt record reaching back 10,000 years.

By gathering additional information about melt rates as well as high-resolution densities in two shallow cores, we developed an approach to correct the annual melt content for the ice volume that gets lost by the thinning process. The result is a melt record with decadal- to centennial- scale resolution for the last two millennia, and the most accurate Holocene climate record for Eastern Greenland so far. The observed changes of annual melt rates show a warm early Holocene, with melt rates higher than in the recent past century, and minimum melting during the Little Ice Age. Current melt rates show a strong increase for

the early 20th century as well as for the time since the end of the past century, with the recent 2012 extreme melting of the Greenland Ice Sheet being the strongest melt event in the past 500 years.

## 1 Introduction

Summer temperatures in Greenland may exceed a critical value which leads to a positive energy balance at the snow surface

and, as a result, causes melting. Through interconnected channels in the pore space the melt water percolates into the snow pack where it refreezes, forming lenses and layers of ice. Higher temperatures will lead to an increased formation of melt water and consequently to an increasing number and thicknesses of melt layers. By analyzing the stratigraphy of the ice sheet with respect to the size and frequency of such melt features it is possible to gain insight into past climate history. As melting generally takes place during Arctic summer season, melt layers can be used as proxies for summer temperatures. Koerner (1977) presented

such a climate record based on melt feature analysis for the Devon Island Ice Cap in the Canadian Arctic, Herron et al. (1981) and Kameda et al. (1995) investigated ice cores from southern Greenland and Alley and Anandakrishnan (1995) did so for central Greenland.





So far, no melt record exists for eastern Greenland, as ice cores are scarce from this region. We now conducted a melt feature study on a core from the Renland ice cap. The ice cap is located on the Renland peninsula and is mostly enclosed by a mountain range. The drill site is located at 2340 m a.s.l. with a mean temperature of -18°C and an annual accumulation rate of 0.49 m water equivalent (w.e.). This is about two times higher compared to the central mainland ( e.g. GRIP has an accumulation rate of 0.23 m w.e./yr (Andersen et al., 2006)). The resulting thick annual layers are beneficial for layer counting and dating, but also lead to faster build up of the load applied to the ice column as overburden pressure increases quickly. The buried layers get compacted and consequently the density increases with depth. Below the firn-ice transition dynamic thinning of the layers takes place in addition. Both processes affect the annual layer sizes and hence need to be considered in order to improve accuracy of the obtained melt record.

In the summer 2015 from May to June, an ice core has been drilled on this remote ice cap (Figure 1) within the frame of the Renland ice cap project (RECAP). It is the second ice core from this location, after a first core has been drilled in 1988 which had a length of 324 m (Johnsen et al., 1992). Both cores reach from the surface down to the bedrock, with the core from 2015 (hereafter referred to as main core) having a length of 584 m. It dates back 120,000 years (Simonsen et al., subm. 2018) and thus contains the whole last glacial cycle and reaches into the Eemian interglacial period (NEEM community members, 2013).

The Renland ice cap, and therefore the RECAP ice core, is exceptional in that it contains bubbles from the top down to the bottom, offering the possibility to study processes that comprise the presence or absence of bubbles (Figure 2a) for the entire ice column. In addition, two shallow firn cores S1 and S2 reaching down to a depth of 71.3 m and 73 m respectively, have been drilled 200 m northwards from the main core with a distance of 30 m between them.

By investigating the differences of the three cores, we examine the lateral variability of melt layers. This allows us to better constrain the overall uncertainty. As a result, we present two melt records, one of high resolution for the past ~2000 years and one of low resolution for the last 10,000 years that provides information about changes in temperature trends of Eastern Greenland back into the early Holocene.

## 2 Data Acquisition: Identifying melt layers visually and using $\mu$CT-density measurements

The three ice cores were cut in the field, each bag of the main core has a length of 0.55 m and those of the firn cores a length of 1 m. Processing was carried out at the Alfred-Wegener-Institute (AWI) in Bremerhaven, which included dielectrical properties and at the Niels Bohr Institute in Copenhagen, where isotope measurements have been conducted.

For this study, two properties of the ice cores were of particular interest: the density and the melt content, both with regard to their changes with depth. In a first step, we obtained the density profiles of the three cores by using the AWI ice core $\mu$CT. It is a computer tomograph specially designed for the scanning of ice core segments with length up to 1 m and is capable of measuring the density profile of an ice core on a sub-millimeter resolution (Freitag et al., 2013). The main core was scanned from 3.3 m until 76 m depth with a resolution of 0.13 mm. At this depth the density reaches 850 kg/m³, indicating the firn-ice transition has been exceeded. S1 and S2 were scanned over their full length (see Tab. 1). As melt layers mainly consist of pure ice with almost no bubbles, they appear as sections of increased density in the $\mu$CT-density profiles. The width of the peak



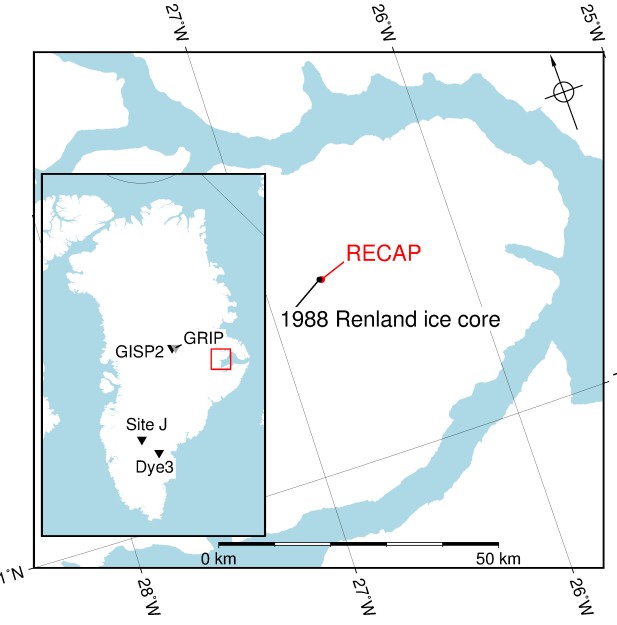

**Figure 1.** The Renland ice cap with the RECAP drill site marked in red and the location where the 1988 ice core has been drilled in black. Red box shows the location of the Renland peninsula. Triangles indicate positions of sites referred to in the main text.

corresponds to the thickness of the respective melt layer. Therefore, the high-resolution density profiles can be used for the identification of possible melt features.

Typical melt layer thicknesses for the ice cores examined in this study are up to 1 cm, but layers probably associated with above-average summer temperatures can be significantly larger (see Figure 3). In general, it is possible to identify such layers

by visual inspection. For this study, we examined the cores S1 and S2 by eye and recorded each position and thickness of the melt layers. Alley and Anandakrishnan (1995) pointed out the problem that for thin layers it becomes difficult to determine whether those are formed by refreezing melt water or other phenomena. For example, wind crusts appear very similar to melt layers. In order to avoid misinterpreting these ambiguous features, we only considered melt layers with thickness $\geq 2$ mm in this study, as crusts are usually not much thicker than 1 mm.

The main core was already cut during standard processing. Therefore we use available line scan images to identify melt layers. A line scan is performed by moving a camera along a microtomed ice core that gets illuminated by an indirect light source (Svensson et al., 2005). In the resulting image, transparent sections of the ice core, such as melt layers, appear in black (Figure 2). Line scan images are available for the whole main core. We examined all images by eye to characterize the contained melt layers. We merged the findings from both methods, $\mu$CT and line scans, to one catalog for each of the three

cores, and double-counted layers were removed manually. Details of the three cores are listed in Table 1.





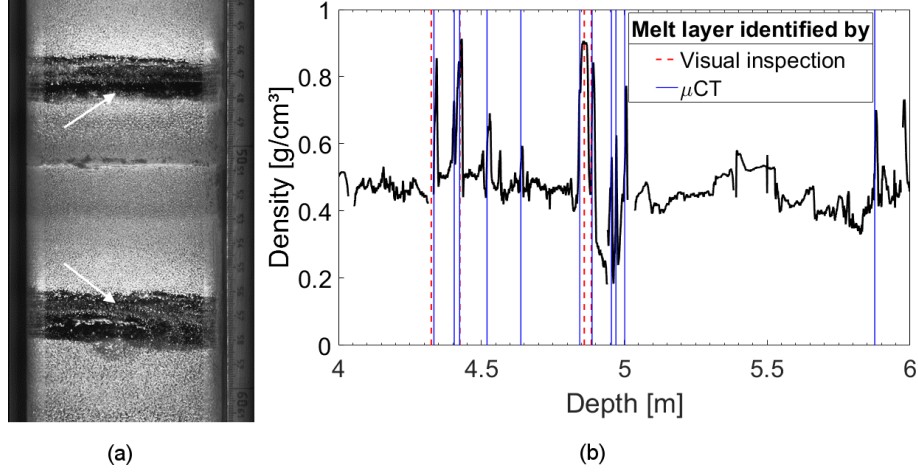

(a)                 (b)

**Figure 2. (a)** Line scan image of RECAP bag143/144. Melt layers can be identified as nearly bubble-free sections (white arrows). Bright line at ruler position 51 cm is the cutting edge between the two bags. **(b)** Density profile of the main core obtained from CT-radioscopy. Vertical lines indicate the identified melt layers.

**Table 1.** Overview of the three cores. In brackets are listed the number of melt layers for the *common depth interval* 3.3 m to 71 m.

| Core | Location | Length [m] | Melt layer |
|------|----------|------------|------------|
| Main | N71°18′13.2″ W26°42′47.2″ | 3.3 - 584 | 1101 (239) |
| S1 | N71°18′15.7″ W26°42′54.0″ | 1.7 - 71.3 | 233 (232) |
| S2 | N71°18′15.1″ W26°42′56.8″ | 1.4 - 73 | 306 (295) |

## 3 Data analysis

### 3.1 Estimation of annual melt ratios

A timescale for the RECAP main core based on $\delta^{18}$O-measurements has been provided by Simonsen et al. (subm. 2018). Using this depth-age relationship it is possible to determine the thickness of the annual layers. For each annual layer we allocate the

5     respective melt layers. The density profiles from the $\mu$ CT allow us to calculate the snow-water equivalent (SWE) for every annual layer. Ideally, melt layers are totally bubble-free (which is rarely the case in reality, compare Figure 2), their respective density will be that of pure ice, which is 917 kg/m$^3$. The fraction of the SWE of the melt layers ($M_{ML}$) and of the total annual layer ($M_{total}$) yields the annual melt ratio (AMR). Assuming a uniform thinning of both layers with age respective depth, the AMR is a thinning-independent quantity. Thus, on an annual time scale AMR values indicate warm summers with a large

10     amounts of melt. On longer, decadal time scales differences in the AMR allow us to distinguish warm and cold periods in the past.





## 3.2 Correction for ice volume loss due to thinning

With increasing depth a load will be applied to the ice column caused by the overburden pressure. It leads to compaction of the annual layers in the firn. Below the firn-ice transition layers will be thinned dynamically in addition as a response to the applied strain (e.g., Nye, 1963; Jansen et al., 2005). Thus the thickness distribution of the melt layers does change with depth.

But below the firn-ice transition melt layers will shrink in the same way as the whole annual layer, i.e. with the same strain rate, and thus thinning does not change the thickness ratio of the layers. The continuous reduction of the layer size leads thin layers to fall below the detection limit of 0.2 cm thickness. Consequently, more and more layers vanish with increasing depth. This means that the AMR values are underestimated the further we go back in time because an uncertain amount of melt layers is not detectable.

An attempt to estimate to which degree the thinning influences the AMR can be made using the thinning rate. Once the thinning rate has been determined, we can calculate at which depth a layer of a certain size will shrink below 0.2 cm. Based on the thickness distributions shown in Figure 3, which are assumed be steady over time, we calculate at which depth each layer of a given thickness will become undetectable to correct the melt volume for the ice volume loss caused by thinning of the layers (or, to be more precisely, the thickness loss since we investigate the reduction of this parameter to estimate the amount

of ice that becomes undetectable).

We derived the thinning rate of the Renland ice cap by using a strain-model modified after Dansgaard and Johnsen (1969), which considers local factors affecting the flow behavior. From radar measurements performed during the pre-site surveys, a valley-structure has been found below the position of the drill site (Johnsen et al., 1992). It is likely that changes in bedrock topography affect the ice flow which also means a changing increase of strain rate with depth. Also, the lowest 47 m have

been identified as dead ice (i.e. ice that has stopped thinning) (Simonsen et al., subm. 2018). Both observations have been incorporated into the thinning model, which results in the depth-dependent strain profile. Further details of the model can be found in the Appendix.

## 4   Results

### 4.1   Common depth interval: 1905 to 2013 CE

The resulting merged catalog lists 1101 melt layers for the main core, 233 for S1 and 306 for S2. In order to compare the three cores, focus has to be put upon the depth sections that are available for all three cores. This *common depth interval* spans from 3.3 m to 71.3 m depth and contains 239 melt layers for the main core, 232 and 295 layers for S1 and S2, respectively. Layer thickness distributions of the cores (Figure 3) show that the main core contains the least amount of melt, about 0.6 times of the summed melt layer thickness that has been found in S2. The melt records of the *common depth interval* further show

variations in time of the distribution of melt between the three cores. Yet we can identify a phase of intensive melt formation at the beginning of the 20th century that appears in all of the three records (Figure 4). The range of this phase is difficult to





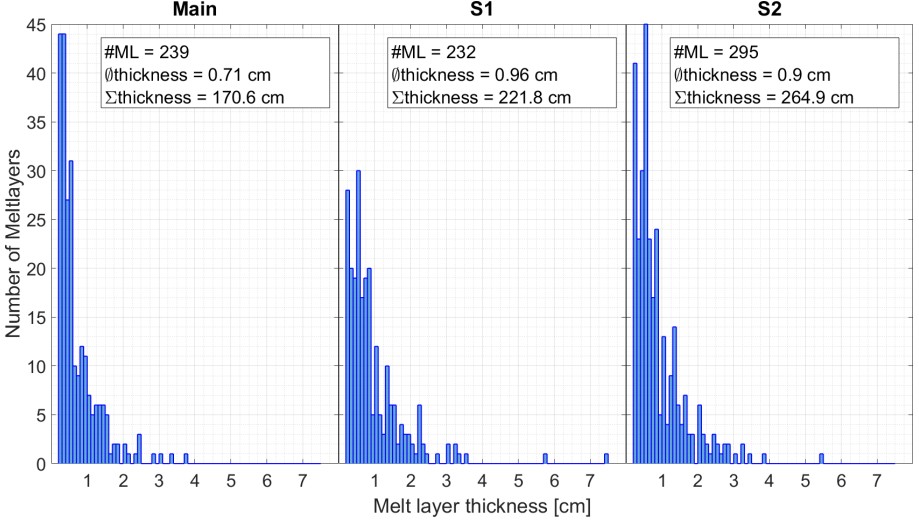

**Figure 3.** Thickness distribution of the melt layer contained in the cores main, S1 and S2 for the *common depth interval* from 3.3 m to 71.3 m depth.

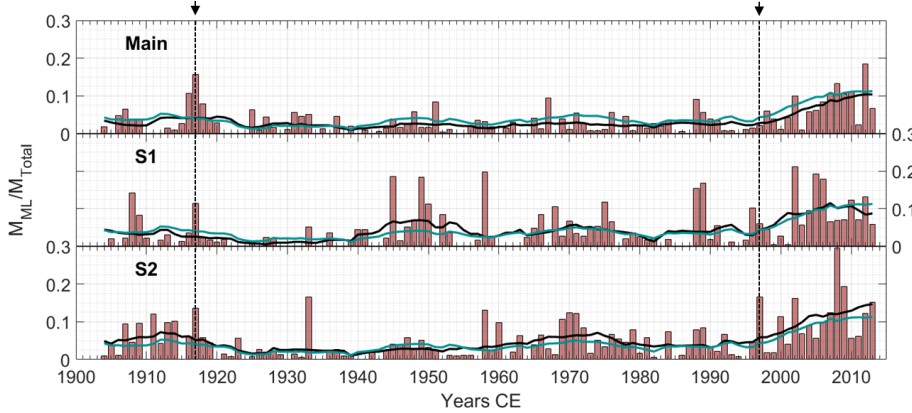

**Figure 4.** Annual melt ratio records of the three cores. The smoothed record is represented by the black line, using a window length of 11 years. Light blue line is the smoothed mean of all three cores. Dashed lines mark the *Early 20th Century Warming* and the onset of the present years warming trend respectively.

determine, but all records share a local peak AMR at 1917 CE and a decay of values soon after it. The mean values begin to distinctly increase in the mid-1990s, a trend that that continues until the end of the record in 2013.

## 4.2 Time period -100 to 2013 years CE

The AMR record of the recent past shows a strong warming trend since about 1860 (Figure 5a, black line). Both the absolute

5 values (red bars) as well as the smoothed curve show a continuous increase which has a first peak in the early 20th century.



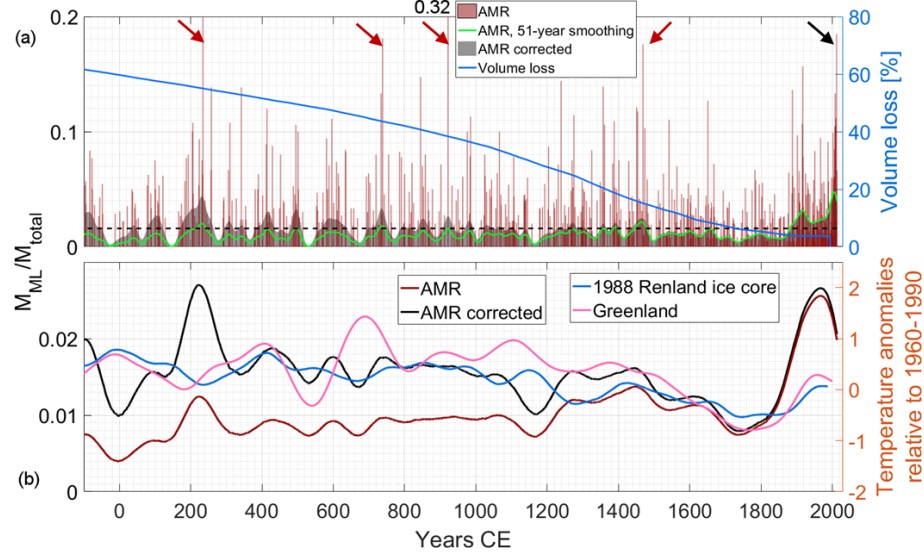

**Figure 5. (a)** Annual melt ratio record of the main core for years 100 BCE - 2013 CE. Blue line describes the percentage of ice volume loss caused by thinning (right axis). Dashed black line shows the mean corrected annual melt ratio, omitting the years with zero melt. Shaded gray areas represents the smoothed AMR curve when corrected for the ice volume loss. Arrows mark prominent melt events, black arrow marks the event of 2012. **(b)** Comparison of the corrected annual melt ratio (black line, left axis) with temperature anomalies obtained from the 1988 Renland ice core $\delta^{18}O$-record (blue line, right axis) (Sonja Wahl, pers. comm., details in the Appendix) and from the reconstructed temperature record for Greenland after Kobashi et al. (2015) (pink line). The temperature anomaly for the 1988 Renland ice core has been calculated relative to the period 1960 to 1988.

From 1990 on the warming trend becomes apparently stronger and the 2012 melting event is prominent (Figure 5a black arrow), when 98.6% of the surface of the Greenland ice sheet has been subjected to melting (Nghiem et al., 2012). This event results in an AMR of 0.18, a value that is reached or even exceeded only four times in the past 2,000 years (red arrows in Figure 5a). Three of these extreme melting events appear between 750 and 1500 CE, which roughly corresponds to the *Medieval Climate*
5   *Anomaly*. Another warming phase is indicated at about 180 to 500. Periods of comparably cool summers with sparse melting can be found between 1680 and 1860, a time of cooling of the northern hemisphere known as *Little Ice Age*, and the years between 0 to 120 seem to have had similarly low summer temperatures. Further back than 2,000 years ago, the volume loss caused by thinning exceeds 60% (compare blue curve in Figure 5a) equaling 115 cm w.e. of ice, and it becomes questionable that the estimated corrections are still valid. As thinning keeps reducing the amount of layers, those layers still detectable just
10  more and more represent the extreme melt events.

### 4.3   Holocene, time period -8,000 to 2,000 CE

The AMR from present until the early Holocene is not corrected for lost ice volume (Figure 6). It can be seen that high AMR values become more frequent compared to the present era for the years before -1,000 CE, where not only the absolute AMR



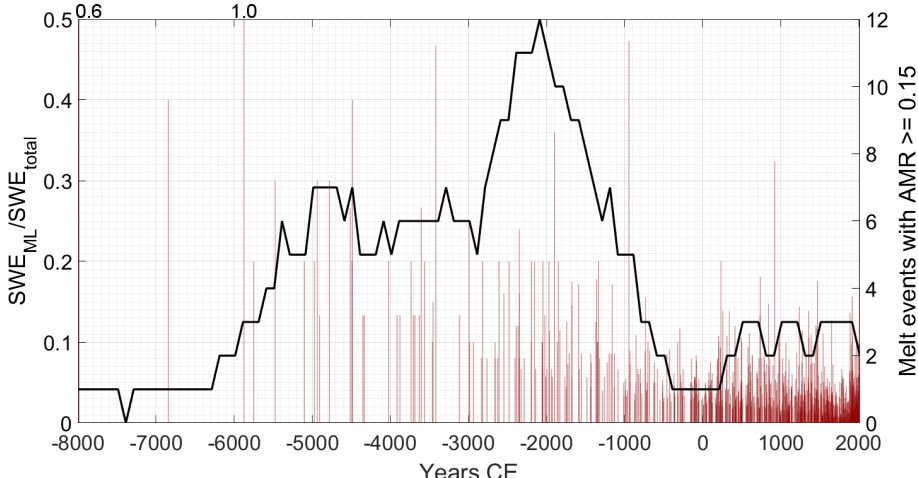

**Figure 6.** Annual melt ratio record of the main core for for the past 10,000 years. The event density (black line) has been calculated using a 1,000 year wide sliding window. Note that no correction for ice volume loss caused by layer thinning has been applied to the annual melt ratios in this figure.

values increase, but also their occurrence frequency rises. In return, the frequency of years containing detectable melt in general decreases further back into the past.

## 5 Discussion

### 5.1 Single-core representativity

Since the lateral distances between the cores are small we assume that weather conditions are equal for all core positions. Also the measured densities show a uniform increase with depth, which indicates that compaction rates are similar in the firn. However, the occurring melt layers show differences in number and size between the three cores (Figure 3).

    As noted in Table 1, S2 shows a significant increase in the amount of melt layers found in the *common depth interval* compared to the other two cores, while the main core and S1 contain about the same number of melt layers. On the other

hand, the layer thicknesses are more similar between S1 and S2 (Figure 3), hence the higher number of melt layers leads to about 40 cm more melt content in S2 compared to S1. This corresponds ti a 20% difference, although the two cores are only 30 m apart from each other. Since the initial amount of melt generated at the surface should be about the same for all core positions, the observed differences can be addressed to spatially varying percolation. Schaller et al. (2018, submitted) conducted a trench study investigating the distribution of melt features in the snow pack on the Greenland ice sheet. The

study observed a concentration of melt along percolation pipes, and that the penetration depth of these pipes varies from some centimeters up to more than 2 m. Similar observations have been made by Heilig et al. (2018) between April and October 2016, using an upward-looking radar on the Greenland ice sheet at 2,120 m a.s.l. height, where climate conditions are comparable to

those of RECAP. The deepest percolation observed by the upGPR is 2.3 m. For RECAP this would correspond to 1.53 years, indicating that melt water percolates into underlying annual layers. Hence, the differences in the melt distribution observed in the three cores is likely expressing the high lateral variability of percolation processes.

The variation of melt content and its distribution is also expressed as varying annual values in the respective AMR records of the three cores. On an annual scale, the deviation can be large; in fact, the annual values rarely match between two cores, and even less when all three cores are compared to each other (Figure 4). This is even the case for the 2012 melt event appearing as the most prominent year in the main core AMR record, but only moderate values are determined in the other two cores. Likely, the varying percolation depth is the cause, as smoothing the record by a three-year window already leads to a similar signal, an indication that irregular melt water redistribution takes place within the first few annual layers below the surface. After applying a 11-year smoothing the signals mostly look the same. This proves our record to have a resolution and representativity on the decadal scale for the *common depth interval*.

## 5.2 Reliability of the AMR record

As described in the data analysis section, a consequence of layer thinning is the resulting underrepresentation of small-size layers that causes gaps in the AMR record as years with former little melt will appear as years with no melting at all. When an annual layer contains multiple melt layers, vanishing of the small layers will reduce the AMR. For example, the AMR of 2012 in the main core consists of 8 individual layers with thicknesses ranging from 0.2 cm to 2.5 cm. According to our thinning rate, the AMR of initially 0.18 for that year will be reduced to 0.13 after about 1,000 years and further reduced to 0.11 after 2,000 years because the thin layers will gradually fall below the detection limit.

Considering that in this way within 2,000 years approximately 60% of the initial melt volume is not detectable anymore (compare blue curve in Figure 5a), we should expect that the AMR values of our record are systematically underestimated the more we go back in time (Figure 5b, red line). The reconstructed lost ice volume takes into account this shortcoming. Without applying the correction the AMR gradually decreases, while the corrected AMR record (Figure 5b, black line) shows a waveform trending around a constant mean. This is in good agreement to the trends of the reconstructed borehole temperature of the 1988 Renland ice core and the general Greenland temperature record by Kobashi et al. (2015) (same Figure, blue and pink line respectively), reconstructed from isotope measurements conducted on the NGRIP and GISP2 ice cores.

The correction for lost ice volume is only valid for those cases when individual layers out of several within an annual layer are lost. In order to correct also the gaps in the record, we need to distinguish whether they are caused by vanishing of melt layers due to thinning, by percolation into underlying annual layers or if indeed no melting took place that year. These requirements go beyond what can be accomplished by the analytical methods applied in the current study.

Our approach ignores the fact that in most parts of the firn, compaction is the dominating process and hence melt layers will not be reduced in their thicknesses contrary to the surrounding layers of softer firn. Instead our model assumes a thinning of all layers starting in the upper part of the firn already. This means the true AMR values in the firn section should be smaller by about 10% maximum when the accumulation rate is considered to be constant in time. Further potential factors that bias the AMR are uncertainties in dating, as the annual layer sizes are calculated from the depth-age relationship. The annual melt



volume, on the other hand, is sensitive to the accuracy of the identified melt layers and their respective thicknesses. Melt layers do not always appear in high contrast to the surrounding firn and ice, the shape can be irregular and in an ice core sometimes a melt layer can't be distinguished from an ice lens. If a melt layer does contain bubbles, then the real SWE value will be lower than the value estimated by using the pure ice density as done in this study. The additional data from density profiles does not fully compensate these problems. We did not quantify the influence of these error sources, but leave it at pointing out that absolute values in the AMR records should require careful investigation. Most of the above mentioned potential error sources intensify their influence with ongoing thinning, hence reliability of the results gradually decreases with age.

## 5.3 Comparison with melt histories from other studies

The AMR record is characterized by a high resolution on a decadal to century scale for the past ~2,000 years, where transitions between different climate phases can be distinguished. The strong warming trend starting in the late 19th century correlates well with the results from Site J in southern Greenland by Kameda et al. (1995). On the contrary, the *Early 20th Century Warming*, found around 1917 CE in our record, appears earlier than in other climate records. Most studies identify this warming phase between 1920 to 1940 CE for the Arctic (Box et al., 2009; Orsi et al., 2017). The onset of the *LIA* around 1500 is also observed by Herron et al. (1981), following the warm Medieval. Not many melt layer studies from Greenland date back beyond this time, and the only other Greenland melt record that reaches back into the early Holocene is the one by Alley and Anandakrishnan (1995) from the GISP2 ice core. They find a strong increase of melt features starting from -2,000 CE and lasting until -6,000 CE, which is about 1,000 years delayed compared to the Renland record. Since the GISP2 record only considers the melt frequency (the number of melt features per 100 years) and is based on much less findings, 1 event per 153 years for the most recent 10,000 years compared to 1 event per 14 years in this study, we believe that the record presented in this study has a higher accuracy and is more reliable. A much warmer than present day Holocene has been indicated in several paleoclimate records, and the observed change of the trend from the warmer to a cooler period at -1000 CE confirms the Northern Hemisphere temperature record of Marcott et al. (2013).

## 6 Conclusions

Melt layers in ice cores are an easy to detect feature with summer temperatures as a most likely cause. It is possible to derive a paleoclimate record by relating the amount of melt to the annual layer thickness and calculate the AMR. In this study, we made an attempt to rule out the biasing influence of random percolation process and dynamic thinning of the ice column on such a melt record.

Our correction approach is based on data from multiple cores from the Renland peninsula and a simple strain model. We present an AMR record corrected for the thinning-induced loss of melt volume for the period -100 CE to 2013 with a decadal to centennial resolution that is in good agreement with climate records from other studies covering this period. The Renland melt record shows a phase of abrupt warming since the beginning of the 20th century, a trend observable in many other climate records of the recent past (Moberg et al., 2005; Semenov and Latif, 2012). Other prominent phases of paleoclimate like the



*Little Ice Age* and the *Medieval Climate Anomaly* can also be well identified. Times of moderate summer temperatures between the years 500 to 700 CE are preceded by another warm phase with melt rates similar to today. Much warmer periods can be found for the early Holocene, which is also validating findings from other climate proxies (Marcott et al., 2013).

Hence, this record proves that melt layers used as indicators of summer temperature can provide climate trends that are confident with other temperature records. The high accuracy of our record makes our method promising for regions with conditions similar to those of the Renland ice cap, with warm summers and near the coast. For regions like East Greenland, where climate records are sparse, melt records can contribute to fill the lack of paleoclimate data.

*Data availability.* The melt layer catalogs of the three cores and the AMR records are available at *doi.pangaea.de/10.1594/PANGAEA.898769*.

## Appendix A: Obtaining the strain rate

This study uses an updated version of the ice flow *kink-model* that incorporates a step rather than a kink and includes a layer of ice, which is frozen to the ground and therefore not moving (Dansgaard and Johnsen, 1969). Instead of a gradual decrease of horizontal flow with depth, the new *step-model* assumes a jump to a reduced flow regime at height $h$ above bedrock. This deviation from the original model is motivated by the local bedrock topography, constellating a valley enclosed by submerged mountains (Johnsen et al., 1992). With these modifications, the age of the ice can be modeled in accordance with the known age reference levels obtained from the GICC05 chronology, which serves as a control method (Rasmussen et al., 2006). Predicting the vertical movement correctly, the flow model can subsequently be incorporated in ice core borehole temperature models and help to reconstruct regional paleotemperatures.

The step model, as seen in Figure A1, is based on the conservation of mass principle. Due to the spatially constrained ice cap extension limits, the height of the ice sheet $H$ is assumed to be constant over the Holocene period. Total gas content measurements support this assumption (Vudayagiri et al.). The driving force is the snow, accumulating on top of the ice cap, which is parameterized as accumulation rate $\lambda_H$.

$$\lambda_H * x = \int_0^H u(z)dz'$$

$$u_H = \frac{\lambda_H}{He} * x$$

$$He = H - h * (1 - f_B)$$

Knowing the horizontal velocities $u(z)$ and utilizing the continuity equation, it is possible to calculate the vertical velocities $w(z)$ close to the ice divide.





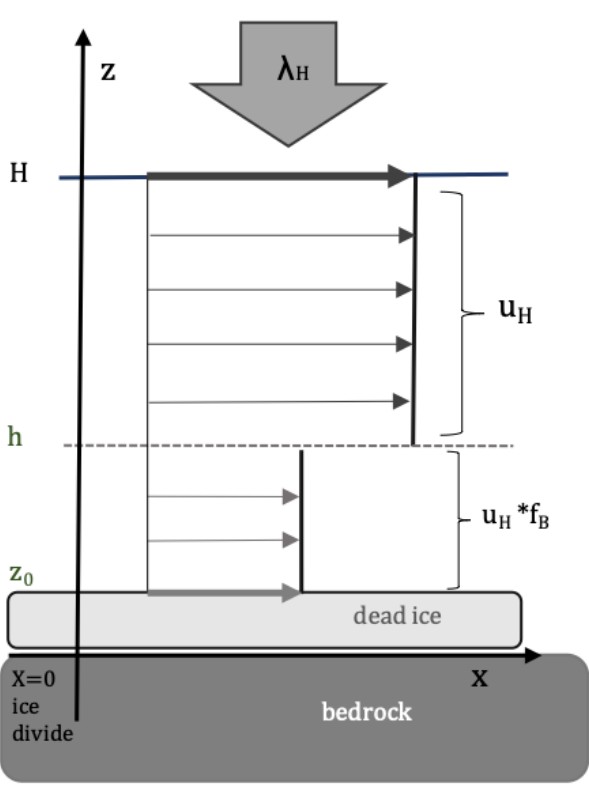

**Figure A1.** The scheme for the ice flow model, including a step at height $h$. The driving force for the ice flow, is the accumulation rate $\lambda_H$.

$$-\frac{du}{dx} = \frac{dw}{dz}$$

$$\int_{w(z_1)}^{w(z_2)} dw = -\int_{z_1}^{z_2} \frac{du}{dx} dz'$$

The model has been tested for accuracy with observations calculating ice age as a function of height above bedrock from the

5  vertical velocities.



$$w(z) = \frac{dz}{dt}$$
$$dt = \frac{dz}{w(z)}$$
$$\int\limits_{t(z_1)}^{t(z_2)} dt = \int\limits_{z_1}^{z_2} \frac{1}{w(z)} dz'$$

The resulting age prediction is compared to known reference age markers of the GICC05 chronology which serves as a validity check.The *step-model* is tunable with four parameters. It has been applied to the RECAP ice core site as well as to the site from 1988. The hereafter presented calculated best fit parameter values are given for both, the RECAP site and for the 1988 Renland drill site (in parenthesis). The calculated best-fit accumulation rates $\lambda_H$ of 0.46 (0.44) cm/year are in agreement with the local observed averaged annual snow accumulation (Johnsen et al., 1992). The corresponding depth of $h$=275 (238) m

at which the models predicts a change from one horizontal flow regime to a reduced flow regime, coincides with the top of the submerged mountain ridge for the RECAP site. The dead ice layer $z_0$ at the bottom of the ice filled valley is about 49 (36) m thick. These layer thicknesses match well with observed annual layer minima in both ice cores (Bo Vinther, pers. comm.). The dead ice is not moving but builds the grounds for the sliding of the layers lying on top. This ice is sliding horizontally at a ratio of $f_B$=0.27 (0.16) compared to the top layer of snow. With the presented parameter values, least-square best fits to the age

markers can be generated and are visualized in Figure A2 a and b. For comparison, the best fit calculations from the *kink-model* are given as well. This *step-model* is the basis for the strain rate, used in this paper.

The flow model was then used to reconstruct paleotemperatures from the measured borehole temperatures of the 1988 borehole following the approach of Paterson and Clarke (1978). The paleothermometer was calibrated by fitting a borehole temperature model outcome to the observed borehole temperature curve. The obtained sensitivity is $dT_S/\delta^{18}O = 1.36°C/‰$

with surface temperature $T_S$. This dependency was used to translate the 1988 Renland ice core isotope record into a temperature curve which is presented in Figure 5 as *1988 Renland ice core*.

*Author contributions.* J.F. carried out the $\mu$CT measurements to obtain the density profiles; Line scan images have been produced by S.K.; B.V. provided the timescale and the step model; The research for this article was supervised by J.F. and O.E.; Data acquisition and analysis performed by T.T.; Manuscript prepared by T.T. with contributions from the co-authors, except for the appendix written by S.W.

*Competing interests.* Author Olaf Eisen is a member of the editorial board of the journal.





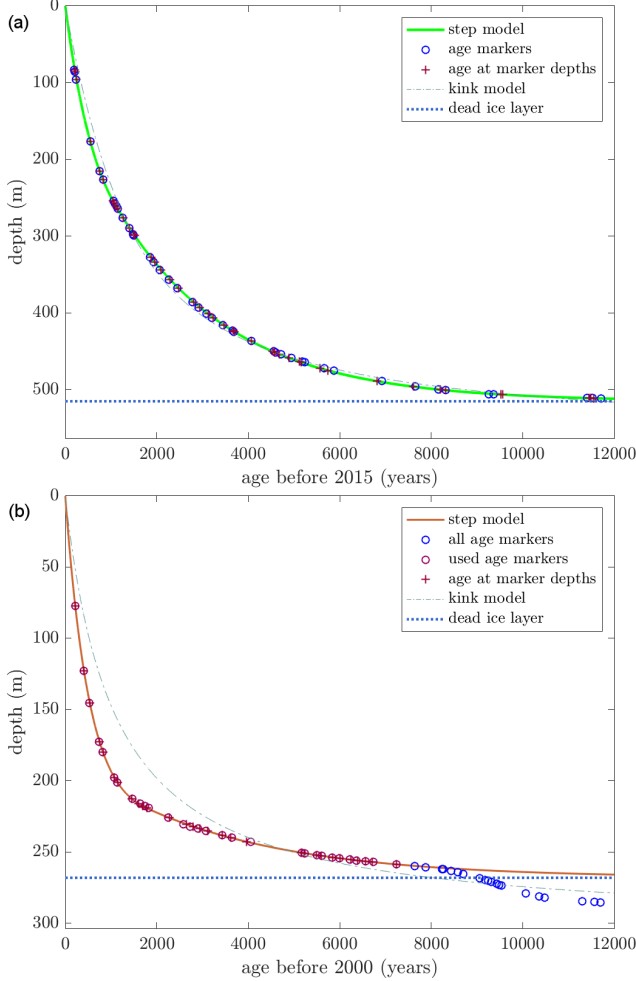

**Figure A2.** The modeled Holocene ice age as a function of height above bedrock is displayed for the **(a)** RECAP core and **(b)** 1988 Renland ice core together with the age markers taken from the GICC05 chronology. The red crosses indicate the calculated age at the reference age levels. The best fit result using the traditional kink-model is displayed as well in gray.

*Acknowledgements.* This study was supported by the Deutsche Forschungsgemeinschaft under grant FE 2527/2-1 RE 3002/3-1. The authors thank the RECAP field team. The RECAP ice coring effort was financed by the Danish Research Council through a Sapere Aude grant, the NSF through the Division of Polar Programs, the Alfred Wegener Institute, and the European Research Council under the European Community's Seventh Framework Programme (FP7/2007-2013) / ERC grant agreement 610055 through the Ice2Ice project and the Early Human Impact project (267696).



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
