# Peer review of "10,000 years of melt history of the 2015 Renland ice core, East Greenland"

_The Cryosphere, 2018_

## Referee Comment (RC1) · Anonymous Referee #1 · 18 May 2019

The authors use ice-core melt records, and melt ratios (AMR) from three recently recovered cores from the Renland Ice Cap (RECAP Project) in East Greenland, in order to develop an independent Holocene temperature record for the region. One of these cores was drilled to bed (534m), while the other two were firn cores drilled approx 200 meters from the primary site (to a depths of ∼71 and ∼73 meters). Using a combination of visual inspection, line scanning, an well as micro-CT imagery, the authors compiled a record of melt layer occurrence and properties, particularly over a shared/common interval between the three cores (representing years 1905 - 2013 CE). Using the compiled AMR record they find climatic trends that agree fairly well with other records for Greenland over similar intervals...specifically the past ∼2000 years. Other specific events like the LIA, MCA are also observable in the results. Smoothing is used to help

account for the effects of percolation

Scientifically, these new data are important and should be published, particularly considering the lack of independent records from this part of Greenland. Over the common interval there is good agreement in the melt-ratio records of the three cores, with several important matching trends from other records. The appendix does a nice job detailing strain rate calculations.

I do think there are some issues with this manuscript however, particularly with regards to it's clarity and the reliability of the record with time. Much is discussed regarding the correction for ice volume loss due to thinning, but I think there needs to be more discussion on this point. For instance, rather than just eliminating all sufficiently thinned layers at depth, why not also try to estimate a crust correction using the shallow cores. . .which could be then applied to the main core so that a blanket correction for thinned layers does not need to be applied as it is (or can at least be validated)

Several of the figures (or their captions) need some clarification as well regarding smoothing intervals, and what's being referenced where in the text.

Section 4.3, and the entirety of the older part of the record for that matter, are not really discussed in this paper aside from the one figure and 3 sentences in this section (or what effect corrections might have on those data). Should this part of the record even be included at all?

And lastly, but very importantly. . ..There were several places in the text where the subject-verb agreement or tenses were incorrect, and many sentences that I simply had trouble following contextually or grammatically. I think the manuscript might benefit from a more-detailed copy editing in some capacity.

Some read-through comments:

Page1

14-16 I think some citations that could be added here for these first few

sentences. . .particularly showing that higher temperatures have been shown to not just increase formation of melt water. . .but increase the thicknesses of melt layers. I'm assuming this is in part because the ice layers formed by refrozen meltwater, act as be barriers to further infiltration and result in thicker layering. . .but should definitely add some citations here. I found a couple that might be helpful (see end of comments)

15 maybe "induces" melting, or "produces" melt 15 Second sentence perhaps could be re-arranged to read a bit better. Maybe "This melt water percolates into the snow pack through the interconnected channels in the pore spaces, where it refreezes and forms lenses and layers of ice."

17-22 Might read better if these two sentences were switched around. Note the seasonality of the melt proxy (and the past studies), and then note for example "Thus by analyzing the size and frequency of melt layers within the ice-core stratigraphy from various sites on the Greenland ice sheet, it is possible to gain some insight into past climate trends for the region".

I also think the sentence listing past studies might read a little better from re-arranging or even simplifying by just stating past studies using ice-core melt features to develop a climate record have been carried out on the Devon Island Ice Cap in the Canadian Arctic, as well as in Southern and Central Greenland (Koerner 1977; Herron et al, 1981; Kameda et al., 1995; Alley and Anandakrishnan, 1995)

16 This sentence might benefit from some clarity. Higher temperatures would certainly lead to an increase production of melt water, but does that necessarily mean an increased number of melt layers in addition to thicknesses? If so. . .then perhaps add some citations. Might it also depend on the geometry of the pore spaces and how well the water can percolate to some degree?

1 remove "so far" and just say "No melt records currently exist for eastern Greenland

due to the scarcity of ice core from this region. . ..” 1 This tense reads a bit awkward. Maybe just state "Here, we conduct a new melt feature study on a recently-drilled ice core from the Renland Ice Cap".

5 "mainland" seems a bit vague. . .might be better here to simply say "two times higher when compared to rates in central Greenland" (as opposed to "mainland". . .just to clarify) 5-7 buried layers always get compacted and density increases with depth regardless of accumulation rate. Might clarify by saying that the buried layers therefore are compacted and sintered at a faster rate. With this said, it also be worth commenting about the combination of this accumulation rate and the mean temperature. Both control grain growth, sintering rates, firn compaction, and therefore pore-close-off depths (e.g. Gow 1969) Later on it's noted that bubbles exist all the way to the bed in this new ice core, so might just add a quick note while discussing pressures, that while overburden pressures increase quickly, it's not enough initiate bubble-to-clathrate transitions. . ..meaning bubbles exist through to the bed.

10 to echo the editor here, I think instances of "present perfect" tenses (i.e. "has been drilled") are not quite correct in this paper. It implies an action that began in the past, but continues through today and into the future. Probably more accurate to say "an ice core was drilled". . ..or even more active voice like "a team of scientists representing the University of Copenhagen and the US National Science Foundation successfully drilled an ice core in on this remote ice cap as part of the REnland ice CAP project (RECAP). 11 Another example of a sentence that might be reworded for clarity. Maybe something simple like "This was the second ice core drilled from this location, following a 324 m core that was drilled in 1988. Both cores were drilled from the surface to bedrock, with the new 2015 core (hereafter referred to as the "main core") having a length of 584 m"

Does the the 2015 "main core" have a more official nomenclature? like like "RE-CAP2015", or RC2015 ? If so, probably best to use that rather than just "main core".

12 Might reword this here to note how this was dated. For example, "Simonsen et al (in

review) recently used a suite of electrical and chemical proxies (or oxygen isotopes. . .or whatever) to develop an initial chronology for the RC2015 core back to ∼120,000 years at the bed (534 m). This depth-scale indicates that the entire previous glacial cycle, as well as the Eemian interglacial period is contained within the RC2015 core (NEEM community members, 2013).

15 As noted above, you hadn't actually said anything about bubbles existing to the bed. Might just clarify this. Also, I assume when you note "absence of bubbles" here, you mean as in melt layers?

18 this sentence is pretty important, but seems out of place in this paragraph where it is. I think that the first sentence of this paragraph could be added to the previous paragraph and the start the last paragraph by stating, "In addition to the main core, two shallow firn cores S1 and S2, of depths 71.3 m and 73 m respectively, were also drilled ∼200 m northwards form the main core site, and ∼30 apart. Here, by investigating the differences of the three cores, we examine the lateral (spatial?) variability. . . ."

24-26. I think this another example of where some copy-editing might help. These sentences are a bit confusing to me. Are bag lengths significant? are they the same as prepared core section lengths? I assume so, but should clarify. Possible suggestion . . . "Recovered cores were processed at the Afred-Wegener-Institute in Bremerhaven, which included in part, measurements of the dielectrial propertites (any other measurements?). Subsequent isotope measurements were later made on the cores at the Niels Bohr Institute in Copenhagen"

27 can you just say, "For this study, we investigate two specific properties of these cores with respect to depth: density and melt content." ?

28 should probably note that the "AWI ice core uCT" is an instrument. "AWI ice micro-ct (uCT) instrument"

32 Ahh..ok... . . .I see you do specify the bubble-free layers you are talking about are

melt layers. This is what I was noting above. Might still say this earlier. Perhaps note why they are typically bubble-free, or at least site a paper explaining this.

Figure 1 is useful and needed, however it might benefit the viewers to make it larger, and to increase the zoom size of the core sites. As it is, the RECAP and 1988 sites appear on top of each other and the scale bar as is, is not incredibly useful.

3 I think this sentence is a bit problematic. What does "probably" mean in this context? Again, it's not necessarily clear to me as the reader that warmer temperatures always correlate to thicker layers.

5 Just say "visually inspected cores S1 and S2" instead of "by eye". 6 I might also add Orsi et al. 2015 here (Differentiating bubble-free layers from melt layers in ice cores using noble gases) 10 was the standard processing what was done at AWI? or was that done in the field? (or at another location?). Were these lines scans done before sending cores to AWI or Copenhagen? it's somewhat unclear when and where this took place. 10-14 Another paragraph that I think could be clarified and/or simplified. change "by eye" to "we visually inspected all images to…" 13 What exactly do you mean by "characterize"? I'm assuming this meant counts, thicknesses, densities, and any other visual properties? Table 1 is just a count, so it's a bit confusing.

3 Ok…I see now that the depth-age scale was based on isotopes. Perhaps note this back in intro.

page 5

7 you note this above in mm, but here and in your figures as cm…might be better to standardize this. In general it seems that this page and its paragraphs do read clearer than previous sections. 13 It's noted that the melt distributions are assumed to be steady over time. I assume what's meant here is that the particular distributions of

existing melt layers are assumed to be steady through time as they progress through depth in the core during burial and thinning? 25 tense is a bit off here. Maybe easier to just say "we focused on an overlapping depth interval shared by all three cores" 28 "Melt-layer" instead of just "Layer". 29 "0.6 times of the summed melt layer thickness" seems confusing. maybe easier just to say only 60 percent the number of layers found in the equivalent section of core S2.

page 6

In figure 4 it might be helpful to put in the y-axis label "AMR" in addition to MML/Mtotal, just for clarity. is this supposed to reference Figure 4 here (not 5a)? If it is supposed to be 5a, then should it be the green line not black?

2 "was subjected to melting" not "has been" same as above regarding figure 5…might put "AMR" in the y-axis label as well 10 probably better to say from the early Holocene to the present (or from the present back through the early Holocene) Figure 5b is great, but shouldn't the shaded grey of subplot a match the black line of subplot b? They are both the AMR record corrected for thinning? Are they smoothed differently and not both 51 years?

The entirety of section 4.3 is only 3 lines, and I while the figure is fairly clear, I think that the accompanying text is confusing. You note that no correction was made for the Early Holocene for ice loss…Wouldn't that affect these results? Especially If a correction is as high as 60% at 2000 yrs BP. There are a lot of data for the Early Holocene, but this is the section and figure showing it…and very briefly. Otherwise, maybe this section should just be dropped completely.

I think the last sentence is supposed to be saying "in contrast, the frequency…" with this said, i think some more explanation is warranted here. Going back in the past, there are more, higher magnitude measured AMR values. Does this mean there was

more warming then, or simply higher magnitude warming events? Also, the lack of measured AMR values in the past is that mostly due to detectability then? I'm not sure this is possible, but is there a way to show a "Corrected" value like you do in figure 5 to account for this?

page 8

11 ti should be "to" 13 not sure if submitted papers should be referenced as "submitted" or "in review". Editor can address this I'm sure.

page 9

27 "correct also" should be "also correct" 25-29 I think this is a really important point here. Is there a different way to maybe account or correct for "crusts" so that all layers thinner than .2 cm can still be counted? and not simply treated as "volume loss" due to thinning?

page 10

15-17 when discussing temporal trends, they should be referenced chronologically. It doesn't make sense to say "starting from -2000 CE and lasting until -6000 CE". This should be written chronologically. This should be addressed throughout the paper as there a few instances of this. 24 "most likely cause" is vague and not necessarily true depending on site conditions. maybe saying a "principal" cause is better.

Pohjola, V. and 7 others. 2002a. Effect of periodic melting on geo- chemical and isotopic signals in an ice core on Lomonosov- fonna, Svalbard. J. Geophys. Res., 107(D4), 4036. (10.1029/ 2000JD000149.)

Opel, T., Fritzsche, D. and Meyer, H., 2013. Eurasian Arctic climate over the past millennium as recorded in the Akademii Nauk ice core (Severnaya Zemlya). Climate of the Past, 9(5), pp.2379-2389.

Winski, D., Osterberg, E., Kreutz, K., Wake, C., Ferris, D., Campbell, S., Baum, M.,

Bailey, A., Birkel, S., Introne, D. and Handley, M., 2018. A 400‐Year Ice Core Melt Layer Record of Summertime Warming in the Alaska Range. Journal of Geophysical Research: Atmospheres, 123(7), pp.3594-3611.
* * *

---

## Referee Comment (RC2) · Anonymous Referee #2 · 24 Jun 2019

General comments:

As the authors point out, melt records are sparse for the Greenland Ice Sheet, and therefore the Renland melt record presented in this paper will be a great contribution to the collection of ice core paleoclimate records. In order to be useful, though, the time spans as well as the claims of shifts in melt/warming trends need backing by statistical analysis.

This manuscript would also be greatly improved by adding more detail to all sections. Below, I have pointed out the most confusing sentences that require more information. Since the manuscript length does not run too long, all sections could be improved by removing all vague statements.

[Figure]

Lastly, the English in this manuscript needs improvement, especially within the discussion and conclusion sections. The authors also need to add transitional words or phrases at the beginnings of sentences, connecting words or phrases (e.g. "and," "as well as," etc.) within many sentences, and commas after the transition words that already exist, in order to make the manuscript more readable. There are also many cases of mixing past and present tenses in sentences, which need to be corrected.

Specific comments:

Abstract: The statement in the final sentence of the abstract is not supported in manuscript below. Please add discussion, where appropriate, to specifically support the claim that the 2012 melting event was the strongest in past 500 years.

Section 2:

1) On Page 3, Line 3: This first sentence of the paragraph is a general description of a result, and therefore is more appropriate for the results section than the methods section here.

Section 3.1:

1) On Page 4, Lines 4-5: What do you mean by 'allocate' here? Do you mean that you count the number of melt layers between the depths of the annual layers?

2) Is the depth-age scale impacted by the frequent formation of melt layers at a site like Renland? If so, how is that incorporated into your error estimate of AMR?

3) It has been shown that large melt events often have multiple ice lenses/melt layers, and that some of those layers penetrate deeper than that year's firn layer (e.g. Nilsson et al., 2015). How do you account for these events depositing a melt layer in the previous year's firn layer? (I see now that you briefly mention this potential source for error in Section 5.2, but I think it's worth mentioning this earlier in the paper here as well for clarity)

Section 3.2:

1) The fourth and fifth sentences in this paragraph contradict each other (Lines 4-6). Consider rewriting this paragraph to make it more clear that there's a difference in the deformation/thinning rates of the firn layers, but that the thinning rate does not affect the thickness rate in the ice layers below the firn-ice transition depth.

2) In the second paragraph, it is unclear how the melt layer thickness distributions aid in calculating the depth at which a layer will thin to below 0.2 cm. Please explain this method of analysis further here.

Section 4.1:

1) What figure shows evidence for the statement "The melt records…further show variations in time of the distribution…"? Also, what does this mean? This is a confusing statement.

2) This would be a good place to describe the meaning of the variables in the legend for Figure 3, and what the values of those variables indicate.

3) Is this a statistically a peak in AMR in 1917? Without any statistics, it's hard to differentiate whether that peak is more significant than the increases in AMR during the 1940s or the 1970s shown in Figure 4. Again, it is hard to compare the recent AMR trend (since the mid-1990s as you state) to 1917 without any statistical analysis.

Section 4.2:

1) There is no black line in Figure 5a, are you referring to the green line in Figure 5a in the first sentence? If so, also correct this labeling within the figure caption.

2) Is the year 1860 when the AMR statistically rises above the background variation in AMR? Statistical analysis to needed to back up this statement.

3) The 1990 warming trend the authors state here isn't visible in the temperature anomaly record shown in Figure 5b, where the blue line (Renland) looks to be at a

plateau and the pink line (Greenland) looks to be decreasing in the 1990s. Please further describe how you found a warming trend at this time.

4) Again, a statistical analysis of the warming and melting trends will allow the authors to state which periods of time in the records were significantly warmer or cooler.

Section 4.3:

1) In the section header, use 8000 BCE instead of -8000 CE to be consistent with the rest of the text.

2) Since the authors corrected the AMR for ice loss in Section 4.2, they should clarify this first sentence of this section that they are now considering the full record without the ice loss correction.

3) Again statistics on the peak of AMR in Figure 6 will significantly help the description of the peak/anomalous years in this record. Otherwise these statements in Section 4.3 are far too vague.

Section 5.1:

1) In the second paragraph, "This corresponds to a 20% difference..." What difference are you referring to here?

2) Again statistics are needed to compare the different smoothing records before making a statement that "This proves our record to have a resolution and representativity on the decadal scale..." Also, representativity is not a word.

Section 5.2:

1) What exactly is the correction used for the corrected AMR curve? Please describe this in greater detail.

Section 5.3:

1) What does the phrase "much less findings" mean on Page 10, Line 18? This sentence here (Lines 16-20) is a run-on sentence and is very hard to follow. Please re-structure.

2) Please provide justification for why you believe that your record "has a higher accuracy and is more reliable," beyond stating that you found more melt layers than Alley and Anandakrishnan (1995) did.

3) On Page 10, Line 21, which "observed change of the trend from the warmer to the cooler period. . ." are you referring to? Please be more specific in these statements.

Section 6:

1) Again, it is hard to accept sentences stating proof of correlating trends in records, or accuracy of this method, without any statistical analysis to back up these claims.

Figure 1: It would be helpful to split the inset map of the entire Greenland Ice Sheet and the zoom in map of Renland into a 2-panel figure, so that the geography of the western portion of Renland is visible.

Figure 2b: Are the blue lines indicating melt layers identified visually through the line scan images, or are they identified because they're spikes in the density record? Also, this panel of Figure 2 is never referenced in the text. Consider referencing it towards the end of the paragraph that ends on Page 3, Line 2.

Figure 3: Please add the description of the three variables found in the legends in the figure caption. Also, in the third panel (for Core S2) is the value for the 4th bar from the left 45, or is this data point cut off? Consider increasing the range of the y-axis to make this clearer.

Figure 4: In the figure caption, the phrase "the onset of the present years warming trend" is imprecise and vague. Please clarify what this is referring to.

Figure 5:

1) In panel 5a, you do not describe what the green line represents in the figure caption.

[Figure]

Also, was does the value '0.32' indicate right next to the legend?

2) In panel 5b, it would be helpful to make both of the axes the same color as the lines of their respective records shown within the figure. To distinguish between the AMR and AMR corrected for the left, and Renland versus Greenland for the right, you can use solid and dashed lines.

3) Lastly, it is very difficult to compare the trends of all of the variable in Figure 5a and 5b. Since you spend some time describing the trends since 1860, consider adding another panel to this figure to enlarge that portion of both plots (1860-present) for easier analysis of the data presented.

Figure 6:

1) Why is the annual melt ratio record denoted as SWE_ML/SWE_total on the y-axis here instead of the M_ML/M_total in Figure 4 and 5?

2) What do the 0.6 and 1.0 values at the top of the figure indicate?

3) What do the red bars indicate in the figure?

4) Which axis relates to which part of the figure?

List of Technical Corrections:

Page 1, Line 5: comma needed after 'rates'

Page 1, Line 17: it should be 'thickness' instead of 'thicknesses'

Page 2, Line 1: "We now conducted" is an awkward phrase because it mixes past and present tense. Consider changing this to "We conducted," or "We now conduct."

Page 2, Line 4: remove space in front of 'e.g.'

Page 2, Line 8: The phrase "in addition" is awkward because it doesn't state what it's in addition to. Consider changing this to "...of the layers also takes place."

[Figure]

Interactive
comment

Page 2, Line 10: Switch the order of this initial phrase to instead say something like "From May to June during the summer of 2015." Also, here's another example of mixing past and present tense. Please correct the verb tense throughout the paper.

Page 2, Line 21: Stay consistent with capitalizing (or not) "Eastern Greenland" throughout the text

Page 2, Line 28: Write out x-ray micro-computed tomography before using the '$\mu$CT' abbreviation for the first time

Page 2, Line 31: This is not clear. At which depth does the core density reach 850 kg/m3?

Page 2, Line 32: It is awkward to start a sentence "S1 and S2..." Consider changing to "The S1 and S2 cores were..."

Page 3, Line 10: The past tense in needed in the second sentence.

Page 4, Line 5: $\mu$CT is abbreviated differently here than above. Keep the abbreviation consistent throughout the manuscript.

Page 4, Lines 6: the sentence is missing 'and' before "their respective"

Page 4, Line 10: should be 'amount' (singular)

Page 5, Line 2: a comma is needed after 'transition'; In addition to what? It is awkward to have the phrase "in addition" without saying what is being compared.

Page 5, Line 5: Remove "but" from the beginning of this sentence

Page 5, Line 10: the phrase should be "the degree to which" instead of "to which degree the"

Page 5, Lines 11-15: This is a run-on sentence. Please reword and break into multiple sentences.

Page 5, Line 25: What does "the resulting merged catalog" refer to? This transition

doesn't make sense without further information.

Page 5, Line 27: "and" or "as well as" needed after "...main core,"

Page 7, Line 5: Be consistent with your use of CE throughout the text.

Page 7, Lines 9-10: This last sentence does not make sense. Please reword.

Page 8, Line 10: "hence" is not the right transition word here.

Page 8, Line 11: "ti" should be "to"

Page 9, Line 10: Should be "an 11-year..."

Page 9, Line 24: restate the figure number instead of saying "same Figure"

Page 10, Line 5: need to add "for" between "compensate" and "these"

Page 10, Line 9: should be "centennial" instead of "century"

Page 10, Line 14: "Period" needed after "Medieval"

Page 10, Line 25: wrong verb tense here again

Page 11, Line 20: the year is missing from this reference

Table 1 Caption: The second sentence is hard to understand. Consider rearranging the sentence to something like "The number of melt layers are listed in..." Also, the number of melt layers are listed in parenthesis in your table, not brackets.